# Comparison between Surgical and Conservative Treatment for Distal Radius Fractures in Patients over 65 Years

**DOI:** 10.3390/jfmk4020026

**Published:** 2019-05-17

**Authors:** Gianluca Testa, Andrea Vescio, Paola Di Masi, Giulio Bruno, Giuseppe Sessa, Vito Pavone

**Affiliations:** Department of General Surgery and Medical Surgical Specialties—Section of Orthopaedic and Traumatology, AOU Policlinico—Vittorio Emanuele, University of Catania, 95123 Catania, Italy

**Keywords:** distal radius fractures, elderly, aging fracture, casting, ORIF

## Abstract

Background: Fractures of the distal radius (DRF) are the most common orthopedic injuries, representing one of the typical fractures indicating underlying osteoporosis. The aim of the study was to compare conservative and surgical treatment, analyzing quality of life and clinical outcome in an over 65 years old population. Methods: Ninety one patients were divided into two groups: the ORIF group (39 patients) underwent surgery, and the conservative group (52 patients) was treated conservatively. The clinical and functional outcomes of all patients were evaluated using Short Form 36 (SF36), Modified Mayo Wrist Score (MMWS), Disability of the Arm Shoulder Hand (DASH), and Visual Analogue Scale (VAS). Range of motion at the joint was measured and compared with the contralateral healthy wrist. Results: No significant difference was found between the overall SF36 score, DASH score, MMWS, and VAS results. Role limitation was significantly better in the surgical group (*p* < 0.05), and complication incidence was significantly higher (*p* < 0.05) in the conservative group. Conclusion: The results of this study conform to recent literature, suggesting that a surgical reconstruction of the radius articular surface in an elderly population provides no clear clinical advantage. Treatment decisions must arise from careful diagnoses of the fracture and communication with the patient.

## 1. Introduction

Fractures of the distal radius (DRF) are the most common orthopedic injuries: one out of every six fractures presented at emergency departments is a distal radius fracture. Almost two thirds of these fractures are displaced and need to be reduced [1]. Epidemiological studies point out that the age rate curve is bimodal, and that the highest incidences are found in children and the elderly [2]. In patients aged 50 years and over, DRF is one of the typical fractures indicating underlying osteoporosis, and US Preventive Services Task Force [3] and EULAR/EFFORT recommendations suggest an additional investigation with dual energy X-ray absorptiometer in these patients [4]. The age-adjusted incidence in large studies ranges from 73 to 202 per 100,000 in men and from 309 to 767 per 100,000 in women. The orthopedic literature features several surgical options for this kind of injury, and each option has its own peculiar advantages and complications. The American Academy of Orthopaedic Surgeons is currently unable to recommend any specific treatment—whether conservative or surgical—and, in the latter case, they also do not suggest which surgical approach is the best one [5]. Choice of treatment depends on many factors, such as the patient’s age, life style, associated medical conditions, compliance, functional demands, limb dominance, type of fracture, severity and alignment of the fracture, condition of the soft tissue, and concomitant fractures [6,7]. Treatment by closed reduction and cast immobilization can be carried out on a large scale at low expense and without admission; however, this often leads to poor radiological results and redisplacement, which can be as high as 40%, according to Mulders et al. [1]. Several surgical options for distal radius fractures have been described, such as percutaneous pinning and casting [8] and external fixation [8,9]; Open Reduction Internal Fixation (ORIF) using the volar locking plate technique represents the most chosen option [9,10]. A recent randomized controlled trial showed that ORIF treated patients have faster recovery of function compared with external fixation, however, no functional advantage was demonstrated at 2 years follow up [11]. 

We compared the two main treatments that are used in the elderly (i.e., conservative and ORIF) and evaluated the clinical outcomes. We then tried to clarify whether or not an open reduction internal fixation is superior to closed reduction and casting.

## 2. Materials and Methods 

### 2.1. Aim of the Study

This study compared conservative and ORIF treatments of distal radius fractures in patients over 65 years old, and analyzed their quality of life and clinical outcomes after the treatment.

### 2.2. Study Design

Between January 2012 and December 2016, 312 patients over 65 years old who were treated at our institute for a DRF were examined retrospectively. The following inclusion criteria were used: (a) age over 65 years; (b) radiologically confirmed and non-exposed distal radius fracture (with or without an associated ulnar styloid process fracture); and (c) absence of other fracture in the upper limb or the hand. The following exclusion criteria were used: (a) open fracture; (b) associated fracture of the scaphoid and/or other carpal bones; (c) pathological fracture. Patient who did not meet the previous criteria or did not complete the follow up program or did not give consent were excluded (*n* = 221). Fractures were classified following AO principles—extra-articular [A], partially articular [B], and articular [C] [12]—based on radiological features and articular involvement. The cohort, composed of 91 patients aged over 65 years, was divided into two groups: the surgically treated ORIF group (39 patients) and the conservatively treated conservative group (52 patients) (Table 1). 

### 2.3. Interventions

All fractures underwent an initial closed reduction; the conservatively treated patients were immobilized with a long cast, although the surgically treated patients were immobilized with a splint. Patients in the conservative group were radiologically examined after 1 week to verify the stability of the reduction; the conservative group patients who showed a loss of reduction after one week had a new reduction of the fracture and/or cast wedging. After 3 weeks, the arm portion of the cast was removed to allow pronosupination movements, while the forearm part was kept for another 3 weeks. After 6 weeks, the cast was removed, and patients were advised for a functional recovery of the wrist articulation. Patients of the surgical group were treated within 1 week from the traumatic event. Surgery was proposed to patients who showed a complex fracture pattern, that is, with post reduction radial shortening > 3 mm, dorsal tilt superior to 10 degrees, or intra-articular displacement or step-off > 2mm [13]. Results were evaluated intraoperatively and postoperatively through radiographs. After the surgery, a splint was applied to the forearm. The immobilization was kept for two weeks and followed by rehabilitation. The rehabilitation consisted of functional training in self-care and home management; functional training in work, community, and leisure; flexibility exercises; application of elastic compression sleeve; massages; and passive joint mobilization. All patients underwent 1, 3, 6, and 12 month clinical and radiographic follow ups.

### 2.4. Clinical Assessment

The clinical and functional outcomes of all patients were evaluated using three questionnaires at the 6 month follow up, and the range of motion (ROM) in injured and contralateral wrists. The Short Form 36 (SF36) was used to analyze quality of life, the Modified Mayo Wrist Score (MMWS) and the Disability of the Arm Shoulder Hand (DASH) were used to assess wrist function, and a Visual Analogue Scale (VAS) was used to evaluate any pain.

#### 2.4.1. Short Form 36 (SF36) and Quality of Life

The SF-36v2 Health Survey [14] is a multipurpose, short-form health survey with 36 questions that yields an eight-scale profile of functional health and well-being, as well as two psychometrically-based physical and mental health summary measures and a preference-based health utility index. It can be used across all adult patient and non-patient populations for several purposes, including screening individual patients, monitoring the results of care, comparing the burden of diseases, and comparing the benefits of different treatments. This survey is composed of 36 questions with standardized answers, and is organized into eight multi-item scales: physical functioning (PF), role limitations due to physical health problems (RP), bodily pain (BP), general health perceptions (GH), vitality (VT), social functioning (SF), role limitations due to emotional problems (RE), and general mental health (MH). All raw scale scores are linearly converted to a 0 to 100 scale: higher scores indicate better well-being and higher functioning of the examined patient [15].

#### 2.4.2. Disability of the Arm, Shoulder and Hand Score (DASH)

The Disability of the Arm, Shoulder and Hand Score (DASH) is a 30 item questionnaire that evaluates symptoms and physical function; each item has 5 responses [16]. The questionnaire contains 30 items: 21 items evaluate difficulty with specific tasks, 5 items evaluate symptoms (2 pain, 1 numbness, 1 stiffness, and 1 weakness), and 1 item evaluates each of the following: social function, work function, sleep, and confidence. The score is scaled between 0 and 100, with higher scores indicating worse upper extremity function. For example, a normal, healthy wrist would score 0, while a maximally disabled one would score 100. The DASH score helps to capture the patient’s perception of their own upper extremity as a single functional unit [16].

#### 2.4.3. Modified Mayo Wrist Score (MMWS)

The Modified Mayo Wrist Score (MMWS) [17] is another instrument used to evaluate wrist function. There is a total of 100 points that are divided among the evaluator’s assessment of pain (25 points), active flexion/extension arc as a percentage of the opposite side (25 points), grip strength as a percentage of the opposite side (25 points), and the ability to return to regular employment or activities (25 points). Pain is rated as none (25 points), mild (20 points), moderate (10 points), or severe (0 points) by the evaluator based on the patient’s subjective description. The total score ranges from 0 to 100 points, with higher scores indicating a better result: an excellent result is defined as 90–100 points, good is defined as 80–89, fair is defined as 65–79 points, and poor is defined as less than 65 points [18]. This score was used to obtain objective data regarding wrist articulation function after treatment by comparing the results with the contralateral healthy limb.

#### 2.4.4. Pain Evaluation VAS (Visual Analogue Scale) 

The pain Visual Analogue Scale (VAS) [19] is a single item scale, a unidimensional measure of pain intensity. For pain intensity, the scale is most commonly anchored by “no pain” (score of 0) and “pain as bad as it could be” or “unbearable pain” (score of 10). The VAS is widely used due to its simplicity and adaptability to a broad range of populations and settings, and is broadly accepted as a generic pain measure.

### 2.5. Radiographic Evaluation

Standard antero-posterior and lateral projections of the wrist joint were taken in the emergency room after cast application, and after one week of immobilization in the conservative cohort.

### 2.6. Statistical Analysis

Continuous data are presented as means and standard deviations, as appropriate. The t-test was used to compare the clinical assessment preoperatively and postoperatively. The chi-square test was used to verify the homogeneity of the two groups based on age, sex, and the injury side and complications. The selected threshold for statistical significance was *p* < 0.05. All statistical analyses were performed using the 2016 GraphPad Software (GraphPad Inc, La Jolla, CA, USA). 

## 3. Results

### 3.1. Sample

A total of 91 distal radius fracture cases of patients older than 65 years were retrospectively examined. The cohort mean follow-up was 14.3 ± 2.3 months (range: 12–17 months). The mean age at trauma was 73 ± 7.3 for the surgical group (range: 65–98) and 76 ± 8.1 for the conservative group (range: 65–95). A remarkable predominance of women was present in both samples: 73% and 75% of the surgical and conservative groups were women, respectively. 

Following the AO principles, in the surgical group, 8 out of 39 had a type A fracture (20.5%), 15 had a type B fracture (38.46%), and 16 had an intraarticular fracture (41%); in the conservative group, 19 out of 52 had a type A fracture (36%), 21 had B fractures (40%), and 12 had C fractures (23%). 

### 3.2. Clinical Assessment

The range of motion (ROM) of the injured wrist was compared to the contralateral healthy wrist: 29 out of 39 of surgically treated patients recovered between 75% and 99% of the ROM compared to the contralateral hand, while in the conservative group, this result was achieved by 33 patients. Degrees of flexion, extension, and radial and ulnar deviation were measured and compared between the two groups (Table 2), and showed no significant difference (*p* = 0.82) between the surgically and the conservatively treated patients.

#### 3.2.1. Short Form 36 (SF36) and Quality of Life

The results show no statistically significant difference between the overall results of the SF-36 questionnaire (Table 2). In the role limitations due to physical health problems (RP) scale, a statistically significant difference (*p* = 0.03) was observed. The physical activity (ORIF group mean 22.15 ± 24.95; conservative group mean 47.27 ± 36.9) (*p* = 0.07) and the social activity scores (ORIF group mean 67.41 ± 25.74; conservative mean 44.09 ± 33.54) (*p* = 0.08) of the SF-36 showed a remarkable difference between the two groups with better results in the surgical sample, but this difference was not significant.

#### 3.2.2. Disability of the Arm, Shoulder and Hand Score (DASH)

No statistically significant differences were found (ORIF group mean 22.15 ± 24.95; conservative group mean 29.39 ± 17.96) (*p* = 0.44).

#### 3.2.3. Modified Mayo Wrist Score (MMWS)

No statistically significant differences were found (ORIF group mean 66.25 ± 20.01; conservative group mean 60.9 ± 18) (*p* = 0.51).

#### 3.2.4. Pain Evaluation VAS (Visual Analogue Scale)

VAS scores show no significant difference between the two cohorts (*p* = 0.60).

### 3.3. Complications

A total of 30 (33%) adverse events were registered among the patients (Table 3): 23 (25.3%) occurred in the conservative group and 7 (7.7%) occurred in the surgical group. 

In the conservative group, 12 (13.2%) patients had loss of reduction within one week from the cast application, and in all cases the patients had a new reduction and casting; 5 (5.5%) reported post-traumatic arthritis (PA); 2 (2.2%) had a deformity after healing; and 2 (2.2%) patients suffered finger edema. Additionally, 2 (2.2%) cases of algodistrofic syndrome were registered.

In the ORIF group, complications were: 3 (3.3%) cases of wrist chronic pain and 2 (2.2%) cases of surgical incision pain. One case (1.1%) required removal of the plate.

## 4. Discussion

Distal radius fractures in the elderly population are very common injuries, but there is no common consensus among orthopedic surgeons for surgery or conservative treatments [13]. For patients over 65 years old, there is no evidence to confirm that one treatment is superior to others. Prevention of this fracture is possible by treating osteoporosis with diet [20] and drugs [21], including vitamin D, calcium, bisphosphonate medications, and recombinant human parathyroid hormone (PTH). Osteoporotic bone presents challenges for both conservative and surgical management of DRF, and goals of care should be discussed with the patient prior to deciding on treatment method [22]. In fact, in patients with decreased bone density, a trabecular bone deterioration was found, especially in early postmenopausal years, and subsequent lower bone strength [23]. Several studies have investigated the functional and radiological outcomes in osteoporotic DRF affected patients; some authors have highlighted that osteoporosis has a negative effect on clinical outcomes in DRFs after ORIF, and suggested that the cause may be the complications related to low bone mass density of the distal radius, including loss of fixation and late displacement [24]; other authors [25], have shown an association between elderly osteoporosis and the loss of initial volar tilt, underlining that ORIF creates a strong construct that prevents displacement, even in patients with low bone density of the distal radius. The wide difference in the financial costs of treatment should be considered when deciding on a treatment option [9]; Shauver et al. [26] estimated that Medicare paid $1458.74 for a closed treatment and $3832.17 for internal fixation. Bruce et al. [27] reported a possible cost saving of $1400 for all patients treated with a cast. Surgical treatment was offered, according to AAOS guidelines [13], after an accurate evaluation of the trauma and the patient’s condition and functional needs. Goals of the surgical treatment included a reduction of radial shortening < 5 mm, radial inclination < 15 degrees, intra-articular step-off (or gap) < 2 mm, and sigmoid notch incongruity < 2 mm. Furthermore, when patients refuse or cannot endure a surgical procedure, cast application and immobilization is the only remaining option. Patients of the surgical group underwent open reduction and internal fixation with a locking plate. In several studies, it has been suggested that there is a high correlation between articular congruity and functional outcome in young, active, and high-functioning patients [28,29]. Malalignment following a distal radial fracture can lead to post-traumatic arthrosis and unsatisfactory function, in addition to deformity and chronic pain in the wrist. Restoring articular congruity and radial length through an ORIF procedure is typically highly recommended for the treatment of young patients. However, there are fewer reports in the literature to support the goal of anatomical restoration of the radius articular surface and length in older populations [30]. Young and Rayan [31] found that after a mean follow-up period of 34 months, functional outcomes, personal satisfaction, and return to activities were independent from radiological outcomes. Gutierrez-Monclus et al. [32] also found that radiological outcomes were not related to articular function after 24 months. 

The aim of this retrospective study was to verify whether or not there is a real advantage in using a surgical procedure to treat an older population of patients. The follow-up included a patient-centered quality-of-life evaluation, along with specific and objective wrist and upper arm scores and ROM measurements. We considered that a distal radius fracture would affect the whole upper extremity, resulting in a temporary or sometimes long-term impairment of physical performance. This concept applies especially to the elderly population, who are more sensitive to a health-related reduction of quality of life and mobility skills. Patients who suffer a DRF usually experience long-term functional impairments that restrict daily activities [15]. Furthermore, several factors can additionally impede from recovery after DRF, such as carpal tunnel syndrome (CTS), complex regional pain syndrome (CRPS), malalignment, and post-traumatic arthritis [15].

Our data showed no significant difference in the patients’ overall outcomes between the two groups; this result is in accordance with recent randomized trials that were conducted on the same topic [5,30]. Regarding quality of life (QoL), there was a significant difference only in the “role limitations due to physical health problems” (RL), with worse scores among the conservative patients. This result is seen in a general health status perspective more than a correlation with the fracture: many of the patients of the conservative group decided to refuse surgery due to bad general health conditions (Parkinson’s disease, Alzheimer’s disease, diabetes, heart failure, and senile dementia were the most common diagnoses in this cohort). This accounts for the gap—even if insignificant—that can be noticed between the SF36 overall scores (mean difference: 13.28), and might justify the significant difference between the RL scores. In fact, the self-perception results of their own bad health conditions led many patients of the conservative cohort to refuse the ORIF procedure.

The biggest randomized trial carried out on DRF in the elderly was unfortunately interrupted due to a low recruitment rate [6]. In line with our data, the ORCHID study showed an insignificant clinical difference between the conservative and the surgical groups after three months of follow-up. Toon et al. [9] compared ORIF with non-operative management of intra articular distal radius fracture, both from a clinical and financial point of view, and concluded that the superior radiographic results obtained in the surgical group did not translate to better functional outcomes in the DASH and MAYO wrist scores, strength, pain score, and range of motion (flexion, extension, and radial deviation). However, some studies report contrasting results. Martinez-Mendez et al. [33] analyzed clinical and radiological results of elderly patients that suffered an articular DRF and found that a better clinical outcome was reached in the surgical group. Essentially, in people older than 65 years, the correlation between restoration of anatomic and articular congruity and clinical benefit is still unclear in the long term. 

The VAS score did not show any significant difference between cohorts. However, the causes of pain in the wrist were different between the two groups. Patients in the surgical group complained about pain at the surgical incision due to device irritation, while among conservatively treated patients, pain was caused by post traumatic arthritis (PA).

We found a significant difference in the incidence of complications: 7.7% in the surgical group and 25.3% in the conservative group. Regardless of the fixation method, surgical management may result in complications. As reported by Yamamoto et al. [34], tendon irritation or tenosynovitis is the second volar locking plate complication (21%) cause and second removal plate reason (14%). Neural injuries may also result from the initial trauma or surgery, or they may develop later because of device-related irritation. Other complications include infection, loss of reduction, intra-articular device placement, compartment syndrome, malunion, and nonunion. Complex regional pain syndrome can develop after the nonsurgical or surgical treatment of DRFs [6]. In our surgical cohort, pain in the wrist (5.5%) was the most common adverse event, compared with chronic wrist pain (3.3%) and pain at the incision (2.2%), followed by carpal tunnel syndrome (CTS) due to device irritation (2.2%) for a total of 6 cases. The CTS case was managed with the removal of the plate and carpal tunnel release exercises. In our conservative cohort, there was a remarkably high rate of secondary displacement (13.2%), especially considering that the cohort included type A fractures. This incidence is in accordance with previous literature. Mulders et al. [1] retrospectively evaluated a group of patients with DRF that were initially treated through closed reduction, and reported a 40% rate of subsequent surgeries due to secondary displacement. Bartl et al. [6] also reported a high rate of conversion from conservative to surgical treatment in the ORCHID study due to displacement of the closed reduction, but this did not negatively influence the final outcome of the “conversion group.”

Post-traumatic wrist arthrosis was the second most common adverse event in the conservative cohort (10%). The literature points out that fractures of the distal end of the radius with persistent articular step-off of the distal radial articular surface have an increased risk of radiocarpal arthritis [34,35]. Articular incongruity like step-off or an increased dorsal tilt cause increased pressure and alterations in the pressure distribution, which can lead to cartilage overload and osteoarthritis during long-term follow-up [33]. Knirk and Jupiter [36] found that radiographic evidence of arthritis developed in 91% of patients with any radiocarpal joint articular incongruity, while only 11% of the patients who healed with a congruent joint demonstrated evidence of arthritis at final follow-up. However, radiographic evidence does not always correspond to clinical symptoms. A study on the same topic conducted by Erhart et al. [37] pointed out that the investigated patients had very low pain levels, although considerable wrist OA was present. Furthermore, there was no correlation between the stage of arthritis and the DASH level. This statement could also justify why although our study found a significantly higher incidence of PA among the conservative patients, it did not find a significant difference in MMWS and DASH scores between the two cohorts.

There were several limitations associated with this study. First, this was a retrospective study that was carried out in a single center and did not contain a consistent surgical sample size. Moreover, many of the patients in the conservative group were advised to undergo a surgical procedure, yet they refused due to their current health conditions (i.e., Parkinson’s, Alzheimer’s disease, cardiovascular diseases).

## 5. Conclusions

It remains unclear whether using ORIF with a plate and screw is the best treatment for distal radius fractures in the elderly. Hence, treatment decisions must arise from careful diagnoses of the fracture as well as two-way communication with the patient to evaluate both physical and psychological health status and the functional requests of the patient. Further, larger and randomized trials are needed to proclaim a gold standard treatment for this common injury that affects the quality of life of the elderly. These trials could help to elaborate an efficient therapeutic algorithm that can aid therapeutic decision making.

## Figures and Tables

**Table 1 jfmk-04-00026-t001:** Demographics. AO = AO classification principles; “A” = extra-articular; “B” = partially articular; “C” = articular.

	ORIF	CONSERVATIVE	*p* Value
**N**	39	52	
**Mean age**	73.2 ± 7.29	76.1 ± 8.13	0.08
**Women**	28	39	0.73
**Men**	11	13	
**AO–A**	8 (20.5%)	19 (36.5%)	0.11
**AO–A1**	2 (5.1%)	8 (15.4%)	
**AO–A2**	4 (10.3%)	5 (9.6%)	
**AO–A3**	2 (5.1%)	6 (11.5%)	
**AO–B**	15 (38.5%)	21 (40.4%)	
**AO–B1**	3 (7.7%)	9 (17.3%)	
**AO–B2**	5 (12.8%)	6 (11.5%)	
**AO–B3**	7 (17.9%)	6 (11.5%)	
**AO–C**	16 (41.0%)	12 (23.1%)	
**AO–C1**	4 (10.3%)	5 (9.6%)	
**AO–C2**	5 (12.8%)	3 (5.8%)	
**AO–C3**	7 (17.9%)	4 (7.7%)	

**Table 2 jfmk-04-00026-t002:** Clinical outcomes.

	ORIF Group	Conservative Group	Mean Difference	*p*
**SF36**	63.36 ± 18.54	50.07 ± 22.65	13.28	0.14
*Physical functioning (PF)*	22.15 ± 24.95	47.27 ±36.90	−25.12	0.07
*Role limitations due to physical health problems (RP)*	64.58 ± 48.21	25 ± 33.54	39.58	0.03
*Bodily pain (BP)*	65.33 ± 23.03	57.54 ± 33.70	7.78	0.52
*General health perceptions (GH)*	41.75 ± 20.05	41.45 ± 9.12	0.29	0.96
*Vitality (VT)*	46.25 ± 9.79	40 ± 12.84	6.25	0.20
*Social functioning (SF)*	67.41 ± 25.74	44.09 ± 33.25	23.32	0.08
*Role limitations due to emotional problems (RE)*	91.66 ± 28.86	87.81 ± 27.10	3.84	0.74
*General mental health (MH)*	63.66 ± 12.92	57.45 ± 22	6.21	0.41
**DASH**	22.15 ± 24.95	29.39 ± 17.96	–7.24	0.44
**MMWS**	66.25 ± 20.01	60.9 ± 18	5.34	0.51
**VAS**	2.91 ± 1.62	2.72 ± 1.78	0.19	0.60
**% ROM**	25–49%	25–50%		0.82
	3	5		
	50–75%	50–75%		
	7	14		
	75–99%	75–99%		
	29	33		
**FLEXION**	71.41 ± 19.36	64.8 ± 20.74	6.61	0.44
**EXTENSION**	72.91 ± 17.3	66.5 ± 17.95	6.41	0.39
**RADIAL DEV**	10.91 ± 2.93	10 ± 2.82	0.91	0.45
**ULNAR DEV**	32 ± 8.86	28.2 ± 7.59	3.8	0.28

**Table 3 jfmk-04-00026-t003:** Complications.

	**ORIF**
Chronic wrist pain	3 (3.3%)
Pain and discomfort at the surgical incision	2 (2.2%)
Carpal tunnel syndrome	1 (1.1%)
Finger edema	1 (1.1%)
Total	7 (7.7%)
	**CONSERVATIVE**
Loss of reduction	12 (13.2%)
Post-traumatic arthritis	5 (5.5%)
Deformity after healing	2 (2.2%)
Algodistrofic syndrome	2 (2.2%)
Fingers edema	2 (2.2%)
TOT	23 (25.3%)

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
