# Peer review of "Comparison between Surgical and Conservative Treatment for Distal Radius Fractures in Patients over 65 Years"

_jfmk, 2019, doi:10.3390/jfmk4020026_

Round 1

Reviewer 1 Report

This study analyzes the clinical outcome and quality of life in elderly patients who experience fractures of the distal radius. As the authors state in their conclusions, a surgical option provides no clear clinical advantage in comparison with a conservatory approach. These findings are very important in clinical practice, because let to choose conservative treatment in very old population with displaced fractures.

They say these fractures could be due to osteoporosis. What about data on bone mineral density and/or mineral and bone metabolism are presented. Because female gender is much more represented, they could have influenced the mean values. What about gender differences?

How many of these patients were on antiosteoporotic drugs?

Author Response

Reviewer 1

This study analyzes the clinical outcome and quality of life in elderly patients who experience fractures of the distal radius. As the authors state in their conclusions, a surgical option provides no clear clinical advantage in comparison with a conservatory approach. These findings are very important in clinical practice, because let to choose conservative treatment in very old population with displaced fractures.

Q1) They say these fractures could be due to osteoporosis. What about data on bone mineral density and/or mineral and bone metabolism are presented.

A1) A new paragraph on bone mineral metabolism was added in discussion (see lines 185-197)

Q2) Because female gender is much more represented, they could have influenced the mean values. What about gender differences?

A2) As reported in literature, women are affected of distal radius fractures approximately 4 times more than men, similarly in our study. No statistically differences (p>0.05) were found in gender, related to questionnaires score (see lines 153-154; 161-162; 167-168; 170-171)

Q3) How many of these patients were on antiosteoporotic drugs?

A3) The antiosteoporotic therapy was not recorded in our series

Reviewer 2 Report

sta et al. present a retrospective analysis of 91 patients with distal radius fractures.
The conservative and operativ group are compared using SF36, MMWS, DASH and VAS.

Introduction:
The considerations regarding "osteoporotic" fractures should also include reflections about fracture liaison service (Ostergaard et al. 2019).

Could you additionally mention the Fixateur externe-treatment (e.g. Hoffmann et al. 2012)?

Perform minor language corrections.

Materials and Methods:
Table 1.: Please give exact p-values, not just >0.05.
Was the "new reduction" after loss of reduction followed by significantly more algodistrofic syndrome?

Results:
Please rearrange the whole section, it is confusing.
Please give subclassification of AO-fractures (AO 23-A1 is quite different to 23-A3).
Complications: Who defined an "adverse event"? And is a haematoma an "adverse event"?
Or a second operation?

Discussion:
Please define a "real advantage" (l 192).
line 225-228: All the pain in conservatively treated patients was caused by posttraumatic arthritis? How was this diagnosed?
Please mention osteoporosis management and radiologic follow-up

Author Response

Introduction

Q1) The considerations regarding "osteoporotic" fractures should also include reflections about fracture liaison service (Ostergaard et al. 2019).

A1) The considerations were added (see lines 185-197)

Q2) Could you additionally mention the Fixateur externe-treatment (e.g. Hoffmann et al. 2012)?

A2) The considerations were added (see lines 46-48)

Materials and Methods:

Q3) Table 1.: Please give exact p-values, not just >0.05.

A3) Exact p-values were added in the table (see lines 397-398; 399)

Q4) Was the "new reduction" after loss of reduction followed by significantly more algodistrofic syndrome?

A4) Only 2 cases of algodistrofic syndrome were reported and did not have a new reduction. The sample is too small to have a statistical correlation. However, several studies have suggested that the incidence of algodistrofic syndrome is correlated with old age, female sex, rheumatoid arthritis (RA), and fracture types (Beerthuizen A. 2012; Zyluk A. 2004)

Results:

Q5) Please rearrange the whole section, it is confusing.

A5) The section was rearranged (see lines 139-171)

Q6) Please give subclassification of AO-fractures (AO 23-A1 is quite different to 23-A3).

A6) An additional statistical analysis were carried out by not statistically significance differences were found (see lines 146-147)

Q7) Complications: Who defined an "adverse event"? And is a haematoma an "adverse event"? Or a second operation?

A7) The adverse events are the ensemble of whole complications recorded. Hematoma was not considered.

Discussion:

Q8) Please define a "real advantage" (l 192).

A8) the word “Real” was replaced with “Functional”.

Q9) line 225-228: All the pain in conservatively treated patients was caused by posttraumatic arthritis? How was this diagnosed?

A9) The diagnosis was performed by senior surgeons according to the functional, radiological outcome and Knirk and Jupiter criteria. As well as in posttraumatic arthritis cases, pain was caused by algodistrofic syndrome and finger edema.

Q10) Please mention osteoporosis management and radiologic follow-up

A10) A osteoporosis paragraph was added in discussion (see lines 185-197; 131-132)

Round 2

Reviewer 2 Report

Thanks for the corrections of the manuscript.

However, some points need to be clarified. The manuscript contains several grammatical and typing errors which need to be rectified by a native speaker.

Line 58-9: Could you give the number of approval from "institutional review board"?

3. Results: Please give headings for each paragraph, including the first one.

Line 46-48:

References are placed wrong (e.g. 9: "Marsh, J.L., Slongo, T.F., Agel, J., et al. Fracture and dislocation classification compendium-2007: 328 Orthopaedic Trauma Association classification, database and outcomes committee. J Orthop Trauma 2007, 329 21(10 Suppl), S1-S133." is not the proposed "recent randomized controlled trial".

Line 144: "in the surgical group, 8 out of 39 had a fracture (20.5%)...

Since the comment was "give subclassification of AO-fractures", why are there no A1-C3 classifications?

Line 154, 163, 166, 169, 172: please give exact p-values (not just p>0.05).

Line 148: Please reformat.

4. Discussion: 

Line 186-196: Did these references somehow affect your study? Please refer to your study. For example: Did you offer these patients FLS? Did you measure their BMD? Or did you use FRAX? See Kanis et al. 2013: "European guidance for the diagnosis and management of osteoporosis in postmenopausal women".

Author Response

Q1) The manuscript contains several grammatical and typing errors which need to be rectified by a native speaker.

A1) Grammatical and typing mistakes have been corrected. Native English editing certificate was uploaded.

Q2) Line 58-9: Could you give the number of approval from "institutional review board"?

A2) Being a retrospective study, no institutional review board approval is needed. The sentence, originated from an old frame, has been removed from the text.

Q3) Results: Please give headings for each paragraph, including the first one.

A3) The headings have been added.

Q4) Line 46-48: References are placed wrong (e.g. 9: "Marsh, J.L., Slongo, T.F., Agel, J., et al. Fracture and dislocation classification compendium-2007: 328 Orthopae dic Trauma Association classification, database and outcomes committee. J Orthop Trauma 2007, 329 21(10 Suppl), S1-S133." is not the proposed "recent randomized controlled trial".

A4) The reference has been corrected.

Q5) Line 144: "in the surgical group, 8 out of 39 had a fracture (20.5%)...

A5) Missed type A was added.

Q6) Since the comment was "give subclassification of AO-fractures", why are there no A1-C3 classifications?

A6) Additional data on AO-fracture subclassification have been added in table 1.

Q7) Line 154, 163, 166, 169, 172: please give exact p-values (not just p>0.05).

A7) Exact p values has been added.

Q8) Line 148: Please reformat.

A8) The paragraph was reformatted

Q9) Discussion:  Line 186-196: Did these references somehow affect your study? Please refer to your study. For example: Did you offer these patients FLS? Did you measure their BMD? Or did you use FRAX? See Kanis et al. 2013: "European guidance for the diagnosis and management of osteoporosis in postmenopausal women".

A9) The study did not have the analysis of osteoporotic patients compared to the non-osteoporotic subjects. No additional data have been collected in osteoporotic patients.

Round 3

Reviewer 2 Report

Thanks for the improvements of the manuscript.
Particularly the p-values and subclassification help the reader to better classify your interesting findings.
The only minor point that should be addressed is that in accordance to US (USPSTF) or UK (NICE) or WHO-guidelines, in patients >65 years with a fracture of the distal radius, osteoporosis screening has to be offered. Please mention that in the discussion or (if you did so) in the introduction.

Author Response

Reviewer: The only minor point that should be addressed is that in accordance to US (USPSTF) or UK (NICE) or WHO-guidelines, in patients >65 years with a fracture of the distal radius, osteoporosis screening has to be offered. Please mention that in the discussion or (if you did so) in the introduction.

Answer: your suggestion was added in the introduction.